# *Mycoplasma ovipneumoniae*: A Most Variable Pathogen

**DOI:** 10.3390/pathogens11121477

**Published:** 2022-12-05

**Authors:** Zinka Maksimović, Maid Rifatbegović, Guido Ruggero Loria, Robin A. J. Nicholas

**Affiliations:** 1Department of Pathobiology and Epidemiology, Veterinary Faculty, University of Sarajevo, Zmaja od Bosne 90, 71000 Sarajevo, Bosnia and Herzegovina; 2Istituto Zooprofilattico Sperimentale della Sicilia, 90129 Palermo, Italy; 3The Oaks, Nutshell Lane, Farnham GU9 0HG, Surrey, UK

**Keywords:** *Mycoplasma ovipneumoniae*, respiratory disease, small ruminants, wild ruminants, mycoplasma variability

## Abstract

*Mycoplasma ovipneumoniae*, a well-established respiratory pathogen of sheep and goats, has gained increased importance recently because of its detection in wild ruminants including members of the *Cervidae* family. Despite its frequent isolation from apparently healthy animals, it is responsible for outbreaks of severe respiratory disease which are often linked to infections with multiple heterologous strains. Furthermore, *M. ovipneumoniae* is characterized by an unusually wide host range, a high degree of phenotypic, biochemical, and genomic heterogeneity, and variable and limited growth in mycoplasma media. A number of mechanisms have been proposed for its pathogenicity, including the production of hydrogen peroxide, reactive oxygen species production, and toxins. It shows wide metabolic activity in vitro, being able to utilize substrates such as glucose, pyruvate, and isopropanol; these patterns can be used to differentiate strains. Treatment of infections in the field is complicated by large variations in the susceptibility of strains to antimicrobials, with many showing high minimum inhibitory concentrations. The lack of commercially available vaccines is probably due to the high cost of developing vaccines for diseases in small ruminants not presently seen as high priority. Multiple strains found in affected sheep and goats may also hamper the development of effective vaccines. This review summarizes the current knowledge and identifies gaps in research on *M. ovipneumoniae*, including its epidemiology in sheep and goats, pathology and clinical presentation, infection in wild ruminants, virulence factors, metabolism, comparative genomics, genotypic variability, phenotypic variability, evolutionary mechanisms, isolation and culture, detection and identification, antimicrobial susceptibility, variations in antimicrobial susceptibility profiles, vaccines, and control.

## 1. Introduction

Since it was first isolated from the lungs of sheep with pulmonary adenomatosis in Scotland in 1963 [1], *Mycoplasma ovipneumoniae* has been frequently found in diseased and apparently healthy respiratory tracts of sheep and goats [2]. In some cases, *M. ovipneumoniae* has been responsible for large financial losses mostly due to poor growth rates and productivity of affected animals [3,4]. Diseases caused by *M. ovipneumoniae* infections are known by a variety of names including atypical pneumonia of sheep, mycoplasma pneumonia, non-progressive (atypical) pneumonia of sheep, chronic bronchopneumonia, chronic, non-progressive pneumonia, and proliferating exudative pneumonia [5,6,7]. In Australia and New Zealand, it is also known as “summer pneumonia” due to an increase in the prevalence of the disease in the hotter weather [5]. A coughing syndrome in the USA and UK, characterized by persistent and long-term coughing, and rectal prolapse, is most likely caused by a combination of *M. ovipneumoniae* and *M. arginini* [8].

Apart from sheep and goats, *M. ovipneumoniae* is detected in a variety of wild ruminants, some of which are endangered [9,10,11,12,13,14,15]. This mycoplasma is characterized by a high level of phenotypic and genomic heterogeneity [16,17,18,19]. It shows variable growth in liquid media [20,21], center-less colonies on solid media containing the usual concentration of agar [22], wide metabolic activity in vitro [23], and variations in susceptibility to antimicrobials [2,8,19,24,25].

Despite the studies so far conducted, the characteristics of *M. ovipneumoniae* and its role in respiratory disease of small ruminants, animal health, and productivity remain largely unknown. In view of the increasing importance of this mycoplasma as measured by a greater number of reports in sheep and goat farms internationally and the relatively recent transmission into threatened wildlife species, we believe it is timely to summarize the current knowledge of *M. ovipneumoniae* to inform control authorities and to identify gaps in research with the aim of improving diagnosis and control measures.

## 2. Epidemiology in Sheep and Goats

*M. ovipneumoniae* is an important pathogen causing respiratory disease in small ruminants worldwide. It is also a common predisposing factor for other bacterial pneumonias, particularly pasteurellosis and viral infections, which may exacerbate the pathological process [26]. Nonetheless, the reason for the presence of this mycoplasma in both apparently healthy and affected animals is still unresolved but clearly indicates other factors are important in disease aetiology (see below). An early study showed that nasal and lung isolates represent similar populations, although nasal strains may differ in their ability to colonize the lungs [27].

*M. ovipneumoniae* is mainly transmitted via the respiratory route following close and repeated contact. Infections generally lead to variable morbidity and low mortality, but the outcome and severity of the disease will depend on the existence of strains of variable virulence, the host response, and the presence of co-infections [2]. For example, multiple strains of *M. ovipneumoniae* have been found within flocks and herds and even in single animals, and these infections may lead to more severe disease [27,28]. Conversely, some disease outbreaks have been attributed to single virulent strains [17]. While it is not always easy to definitively link *M. ovipneumoniae* with a severe respiratory disease because of the detection of more rapidly growing bacteria, several reports have provided substantial evidence of mycoplasma involvement particularly in goats [7,29,30].

Its role in pneumonia is often overlooked because mycoplasma diagnosis is not routinely carried out in all diagnostic laboratories [26]. However, an increase in cases has been reported recently by several countries that perform regular monitoring, including Sweden [31], USA [4], France [19], and UK [32]. In 2001, *M. ovipneumoniae* was detected in 88% of 453 domestic sheep operations across USA [4]; in France, between 2007 and 2019, 16.4% of *Mycoplasma* spp. isolates from small ruminants were identified as *M. ovipneumoniae* [19], and in England and Wales, between 2005–2019, *M. ovipneumoniae* represented over half of all mycoplasmas isolated in small ruminants [32]. A random survey of sheep between 2012 and 2014 in six regions of China revealed an average seroprevalence of 18% and PCR detection rate of 10% in nasal swabs and nearly 30% in lungs [33]. A few years later, just over 40% of nasal swabs from sheep in Xinjiang, a major sheep breeding region, were PCR positive [34]. However, the true prevalence of *M. ovipneumoniae* infection will not be known until large-scale studies are conducted in more countries with the establishment of global surveillance systems.

Unlike some mycoplasma diseases such as bovine mycoplasmoses, the prevalence seems to be higher in operations with more open or “herded” practices, where animals are maintained on any unfenced acreage, than in those with more intensive husbandry [4]; this seems counter-intuitive given the nature of most mycoplasma diseases where infection is spread by close contact. For example, in Bosnia and Herzegovina [18,35] and the Republic of Kosovo [30] where small ruminants are managed extensively, severe respiratory disease is widespread. The possible reasons for this include the lack of control of animal movements and the introduction of newly acquired, apparently healthy, and often older animals into a herd of mainly younger stock, resulting in the mixing of infected with uninfected animals [2,30]. Interestingly, in contrast to goats, *M. ovipneumoniae* is more commonly found in younger rather than older sheep [19,36], and the frequency of its detection appears to vary among sheep breeds [34,36]. Higher temperatures and lower relative humidity also may favor the presence of *M. ovpineumoniae* [36,37]. Other predisposing factors contributing to the disease included: high stocking densities and low ventilation rates in intensely bred lambs [38]; stress induced by live animal movements [5], different ecoclimatic factors and season of the year [7,30,39]. A major concern is that once introduced into a herd, *M. ovipneumoniae* may persist for a long period and become difficult to eliminate [8]. In all, increases in the occurrence of *M. ovpineumoniae* appear to be influenced by a range of factors favorable to the spread of the mycoplasma which need to be considered when assessing risk factors.

## 3. Pathology and Clinical Presentation

Clinical signs may be mild and some animals can recover within a few weeks, although in other cases, pneumonia can persist for much longer. During times of stress, pneumonia may be more severe, and infections may result in acute fibrinous pneumonia, consolidated lung lesions, pulmonary abscesses, pleurisy, and even death. Lesions begin with dull red ventral areas of collapse, which are accompanied by bronchiolitis in associated airways. They progress to firm red-grey areas of consolidation over 2–3 weeks, but may continue as grey areas of consolidation, often with attached localized pleural adhesions [5]. Clinical features include increased respiratory rates, coughing, nasal discharge, temperature rise, depression of appetite and growth rate, and a drop in milk yield in ewes [40]. In addition to the lung, trachea, and nose, *M. ovipneumoniae* has been isolated occasionally from eyes and udders, although its role in disease in these sites is unclear [1].

Attempts to reproduce the disease experimentally have been variable, showing effects ranging from pulmonary colonization of lambs without pathological changes, interstitial and lymphoid cell proliferation pneumonias to a proliferative exudative pneumonia [41]. Similar inconsistent results were also reported in experimentally infected young goats [42,43,44]. Experimental mixed infections of *M. ovipneumoniae* with other respiratory pathogens, such as *Mannheimia haemolytica*, have led to more severe disease [45,46]. There have been few studies looking at the potential role of viruses in the development of *M. ovipneumoniae* disease. In a more recent study [47], no obvious clinical signs, evidence of lung damage, or inflammation were observed in any experimentally infected lambs. However, it was suggested that *M. ovipneumoniae* may cause asymptomatic colonization of the upper respiratory tract and impair lamb growth and productivity even in the absence of overt respiratory disease [3,47]. The variations in reported pathological changes and clinical appearance of the disease could be explained by different experimental protocols, variations in virulence amongst isolates, and variable host susceptibility.

## 4. Infection in Wild Ruminants

Until recently, most studies focused on *M. ovipneumoniae* infections in sheep and goats [35,43], and for many years, the host range was believed to be restricted to these animals [2]. However, mainly over the last decade, an increasing number of reports described cases of *M. ovipneumoniae* infections in wild ruminants. First reports were in animals genetically close to domestic sheep (genus *Ovis*), including the thinhorn (*Ovis canadensis nelson*) [9] and bighorn sheep (*Ovis canadensis canadensis*) [10]. Severe infections with clinical signs, similar to those in domestic species, were probably first recorded in bighorn sheep almost a century ago, though *M. ovipneumoniae* had not yet been isolated. Outbreaks to date have so far been attributed to single strains which exhibit a high degree of variability [48,49]. The concern the disease was having on this threatened species fueled the many studies conducted recently to detect the primary cause and risk factors of this bronchopneumonia. Several domestic-wild sheep commingling experiments have concluded that domestic sheep and goats are the primary sources of *M. ovipneumoniae* infections in bighorn sheep [49]. The resultant pneumonia is seen in all age groups, causing considerable variation in severity, with mortality rates typically between 5–100% [49,50]. In addition to coughing and nasal discharge in the free-living animals, lethargy, fever, drooping ears, and head shaking were observed in bighorn sheep experimentally infected with *M. ovipneumoniae* [10,51].

*M. ovipneumoniae* has also been detected in other wild *Caprinae*, including Dall sheep (*Ovis dalli dalli*) showing signs of severe respiratory disease [9], in pneumonic mountain goats (*Oreamnos americanus*) [14], and in desert bighorn sheep (*Ovis canadensis nelsoni*) during an outbreak of respiratory disease [14]. In addition, *M. ovipneumoniae* was identified as a primary cause of highly fatal pneumonia in the muskox (*Ovibos moschatus*), a bovid species belonging to the subfamily *Caprinae*, during two outbreaks with mortality rates of 25–30% [11]. In the cases of respiratory disease in captive Dall sheep and muskox, it was evident that animals had been in contact with domestic sheep flocks from which *M. ovipneumoniae* was also isolated, thus, implicating the latter as the most likely source of infection [9,11].

It is highly likely that an outbreak of bronchopneumonia in mountain goats was caused by exposure to bighorn sheep experiencing a pneumonia epizootic as the *M. ovipneumoniae* isolate recovered from the mountain goats shared the same genotype detected in the sympatric bighorn sheep [14]. Evidence that bighorn sheep may serve as a source of *M. ovipneumoniae* infection for other animals was also supported during an epizootic disease outbreak in which the mycoplasma was also detected in pneumonic cattle (*Bos taurus*) which were in close contact [52].

In recent years, *M. ovipneumoniae* has been isolated from several animal species other than the subfamily *Caprinae*, family *Bovidae* (Table 1).

Highland et al. [12] detected *M. ovipneumoniae* in nasal swabs of six of 230 moose (*Alces alces*), five of 243 caribou (*Rangifer tarandus*) and two of five mule deer. One mule deer had nasal discharge, while other animals had no clinical signs of respiratory infection at the time of sampling. One white-tailed deer (*Odocoileus virginianus*) positive for *M. ovipneumoniae* died during a pneumonia outbreak at a captive facility. Similarly, Rovani et al. [13] detected *M. ovipneumoniae* in a pneumonic lung from a dead free-ranging barren-ground caribou (*Rangifer tarandus granti*) in Alaska. The findings of *M. ovipneumoniae* in a variety of wild ruminants indicate that there may be increasingly closer contact between wild and domestic species, possibly as domestic species encroach on wilder pasture. Alternatively, it could suggest that this mycoplasma has a very wide host range which is only just being recognized.

## 5. Virulence Factors

Our knowledge about the virulence factors and pathogenicity of *M. ovipneumoniae* is slowly becoming clearer. Three genes, *glpF*, *glpK*, and *glpD*, involved in glycerol import and production of hydrogen peroxide and reactive oxygen species (ROS), have been identified, as well as two proteins with a high level of amino acid sequence similarity to bacterial toxins, hemolysin A (*hlyA*) and hemolysin C (*hlyC*) [53]. Various mechanisms that contribute to virulence have been proposed and include alterations in macrophage activity [54], induction of the production of autoantibodies to the ciliary antigen [55], high levels of hydrogen peroxide production, leading to tissue damage [56], induction of cilio stasis, which is correlated with hydrogen peroxide production [57], suppression of lymphocyte activity [58], and production of a capsular polysaccharide (CPS), which facilitates adherence of the organism to ciliated epithelium [59]. Whether all or just some of these mechanisms function during the disease process is yet to be determined.

During in vitro studies, *M. ovipneumoniae* was found intracellularly, as well as on the host cell surface [60]. CPS may be a key virulence component of *M. ovipneumoniae* and may be involved in the inflammatory response [61]. It can induce caspase-dependent apoptosis via both the intrinsic and extrinsic apoptotic pathways, in sheep bronchial epithelial cells, which is mainly attributed to the ROS-dependent JNK and p38 mitogen-activated protein kinase (MAPK) signaling pathways [62]. *M. ovipneumoniae* shows very limited biofilm formation in vitro, which may be because of its polysaccharide capsule [63]. Mechanisms promoting high-frequency surface variations, which have been identified in some ruminant mycoplasmas and are thought to enable avoidance of host defenses [64,65] have not yet been described in *M. ovipneumoniae*.

## 6. Metabolism

*M. ovipneumoniae* ferments glucose, reduces tetrazolium when grown anaerobically, and shows variable tetrazolium reduction under aerobic conditions [23]. It does not hydrolyze arginine, show phosphatase activity, digest casein, nor produce film and spots in the medium. However, it exhibits a wide and varied metabolic activity, particularly in substrate utilization patterns [23]. All strains are able to utilize glucose and pyruvate [66] as well as N-acetylglucosamine, glycerol, and isopropanol [23]. Fructose, maltose, and 2-oxybutyrate are metabolized by most strains, approximately half metabolize mannose, but only one strain metabolized lactate [23]. *M. ovipneumoniae* is able to produce high levels of hydrogen peroxide following oxidative metabolism of nicotinamide adenine dinucleotide hydrogen (NADH) and/or α-glycerol phosphate (αGP), compared with other mycoplasma species [2]. While metabolic profiling of *Mycoplasma* species is generally not very useful in characterizing strains because of their limited metabolic diversity, *M. ovipneumoniae* is an exception and further studies are needed to explore its value in better understanding the diseases it causes.

## 7. Comparative Genomics

Although *M. ovipneumoniae* was first isolated almost 60 years ago [1], there is still much to know. Based on 16S rRNA and *rpoB* gene sequence data, *M. ovipneumoniae* clusters with the mycoplasmas belonging to the hyopneumoniae group, within the neurolyticum/hyopneumoniae cluster [67]. Based on analysis of the 16S rRNA, and 70-kDa heat shock protein (Hsp70) gene sequences, and whole genome sequencing (WGS), *M. ovipneumoniae* is most closely related to the pig pathogen, *M. hyopneumoniae* [68,69,70,71]. The first complete genome sequence of *M. ovipneumoniae*, strain SC01, which was isolated from a goat in China, was reported in 2011 [53]. The goat strain SC01 possesses at least eight coding sequences (CDSs) encoding proteins homologous to adhesin-like proteins of *M. hyopneumoniae*. They include three homologues of a P102-like protein, two homologues of an adhesin-like protein P146, and one homologue each of a P97-like protein, an adhesin, and a P76 protein [53]. Another feature of the SC01 genome is the frequent usage of UUG as initiation codon (21.6%) [53], while in other mycoplasmas, it is generally only used in about 5% of CDSs. Another Chinese strain, NM2010, which was isolated from a sheep in 2014, shared only 728 genes with strain SC01 [72]. In comparison with goat strain SC01, sheep strain NM2010 had one less homologue of the P102-like protein, lacked the P97-like protein, had two additional homologues of the P102 adhesin protein, and had one homologue of the P60 adhesin-like protein [72]. A third genome of a New Zealand sheep isolate of *M. ovipneumoniae*, strain 90, was sequenced and published as a draft in 2020 (Table 2) [73,74].

Currently there are 17 genome entries for *M. ovipneumoniae* in GenBank of the National Center for Biotechnology Information (NCBI) [74]. Based on genome sequencing, most goat strains seem to be distinct from sheep strains (Figure 1) [75].

## 8. Genotypic Variability

Worldwide, *M. ovipneumoniae* isolates exhibit a high level of genetic, protein, and antigenic heterogeneity [16,17,18,76,77]. In early studies on the genotypic characteristics of *M. ovipneumoniae*, the high genetic diversity of New Zealand isolates was demonstrated using restriction endonuclease cleavage analysis of genomic DNA [76,78]. Strains that initially colonized lambs were replaced by different strains during the course of an infection [27]. Although the nasal and lung isolates are not completely distinct populations, the former may differ in their ability to colonize the lungs [27]. Multiple strains can be found within herds or within a single animal, and infection with multiple strains may lead to more severe disease [17,27,28]. The number of *M. ovipneumoniae* strains present in the lungs of the sheep correlates with the severity of the lesions in that the less severe lesions contained fewer i.e., two strains, whereas the more severe lesions contain at least three or, in most cases, four strains of *M. ovipneumoniae* [28].

While the evidence, described above, shows that the concentration of mycoplasmas and number of strains present in the lung are factors in the extent of disease seen in affected animals, differences in virulence between isolates have not been reported other than anecdotal reports from the field; this is due to the fact that properly controlled experimental infections have not yet been carried out. Such experiments may show the existence of *M. ovipneumoniae* strains of varied virulence, which may account for the presence of mycoplasma in both apparently healthy and affected animals, and the variable outcome of the infection.

High genetic variability between strains and the presence of genetically distinct isolates on the same farm have been reported in the UK. Among the 43 isolates analyzed using the random amplified polymorphic DNA (RAPD) method and pulsed field gel electrophoresis (PFGE), 40 RAPD Hum-1, 41 RAPD Hum-4, and 40 PFGE profiles were detected [17]. There was no relationship between the profiles of sheep isolates from different herds and their geographical origins, probably due to extensive sheep movements within the UK [17]. However, isolates associated with the same disease outbreak were found within all groupings, with strains isolated from distant locations having similar profiles [17]. In the USA, Harvey et al. [77], using single primer arbitrarily primed polymerase chain reaction (AP-PCR) and amplified-fragment length polymorphism (AFLP) methods, detected multiple strains within individual animals. As in a previous investigation in the UK [17], it was demonstrated that, in several herds, one strain of *M. ovipneumoniae* could be found in several sheep, although the strains were herd-specific [77]. Subsequent investigations have reported a high degree of genomic heterogeneity among *M. ovipneumoniae* strains in different countries [19,60,79,80]. Multiple strains were detected in goat herds with severe respiratory disease, although differences in genotypic profiles were also seen in strains isolated from clinically healthy goats [18]. Genotypic differences between goat and sheep strains have been shown by RAPD and Hsp70 gene sequence comparisons, revealing a link between strains and host ruminant species [18,19]. Moreover, most European strains isolated from sheep and goats were found to be distinct from Asian strains [18]. Whole-genome single-nucleotide-polymorphism (SNP) phylogeny demonstrated clustering of strains based on their host species, regardless of their geographical origin, and suggested co-evolution of *M. ovipneumoniae* with its host species [19]. Recent studies, using multi-locus sequence typing (MLST), in the USA, also showed that domestic goat strains were distinct from those in domestic sheep [81,82], which had high genetic diversity. However, strains from both domestic ruminant species were found in bighorn sheep [82].

## 9. Phenotypic Variability

The high genetic variability of *M. ovipneumoniae* is reflected in its phenotypic heterogeneity. An early protein characterization study of *M. ovipneumoniae* using SDS-PAGE, conducted in New Zealand in 1985, revealed unique protein profiles for each of eight isolates [78]. Protein heterogeneity of isolates from New Zealand, as well as those from Scotland, was confirmed a few years later [16,76]. Highly variable protein expression was also demonstrated for UK isolates [17]. The number of polypeptides detected by SDS-PAGE varies from 30 to 50, with molecular masses (Mr) ranging from approximately 30 to 250 kDa [16,18,66]. Most of the main proteins were identified as antigenic by immunoblotting analysis, revealing extensive heterogeneity of *M. ovipneumoniae* isolates [16,17,18]. The major conserved antigens of Scottish isolates were 32, 33, 35, 44, 56 (60), 76, 89, 105, and 129 kDa [16]. In a later study in the UK, Lin et al. [66] reported predominant antigens of Mr 30, 40, and 64 kDa (±3 kDa), while most of the serum samples of naturally infected sheep reacted with a 71 kDa antigen. In a previous study also conducted in the UK [17], an antigen of between 40 and 43 kDa was observed within most isolates. Antigens of Mr 26, 40, 42, 65, 83, and 105 kDa were detected in Chinese isolates of *M. ovipneumoniae*, with the 40 kDa and 83 kDa antigens conserved in all isolates [79]. Most of these studies have only been conducted on sheep isolates [16,17,66]. In a recent study, although considerable antigenic variability was detected between isolates, predominant antigens of Mr 36, 38, 40, and 70 kDa (±3 kDa) were found to be common to both sheep and goat isolates, regardless of sample type (nasal swabs or lung), clinical status of the source animal (whether affected or asymptomatic), or geographical origin [18].

This worldwide antigenic variability of *M. ovipneumoniae* isolates complicates efforts to develop vaccines and serological diagnostic tests. However, one of the most conserved proteins of *Mycoplasma* species, the Hsp70 protein, was found to be immunogenic in natural infections of *M. ovipneumoniae*, and thus may be a relevant antigen for vaccine development [69]. Another membrane-associated protein on *M. ovipneumoniae*, elongation factor Tu (EF-Tu), has been suggested as a potential candidate for vaccine development [83].

## 10. Evolutionary Mechanisms

The reasons for a high degree of phenotypic and genomic heterogeneity of *M. ovipneumoniae* have been speculated. Parham et al. [17] have suggested that the presence of extensive variation in *M. ovipneumoniae* results in a large gene pool, allowing the population, as a whole, to respond to changes in its environment, thus providing protection for the species. Under specific selection pressures, an increasing number of isolates adapted to the new conditions are more likely to survive [17]. Perhaps the heterogeneity of *M. ovipneumoniae* could be the result of divergent evolution through horizontal gene transfer of DNA and/or vertical gene transfer after natural selection of new mutations. In contrast to *M. ovipneumoniae*, *M. capricolum* subsp. *capripneumoniae*, a highly pathogenic small ruminant mycoplasma, lacks genetic variation. This has been shown by large-scale genomic analysis to be due to its recent emergence as a species, perhaps less than 300 years ago [84]. This may also account for its virulence in a host to which it is poorly adapted. The extreme heterogeneity of *M. ovipneumoniae* would suggest a longer evolutionary descent, resulting in variable pathogenicity in domestic sheep and goats, to which it has adapted. Its recent introduction into new hosts, like the bighorn sheep and musk ox, may explain its enhanced virulence. However, evolutionary mechanisms in *M. ovipneumoniae* have yet to be investigated.

## 11. Isolation and Culture

While new rapid and reliable techniques are available for the detection of *M. ovipneumoniae*, culture remains an important tool, being an essential step for the characterization of strains, antimicrobial susceptibility testing, and the development of vaccines and serological tests [20,85].

Although specialist transport media for mycoplasmas can be used, some authors prefer the use of the same medium for both transport and culture [30,86]. While relatively easily to isolate, *M. ovipneumoniae* is difficult and time consuming to grow and maintain in culture [87,88]. Growth media usually used for *M. ovipneumoniae* include Eaton’s (PPLO broth base) [89,90], Hayflick’s [87,91], and SP4 medium [92,93], as well as various modifications of these media. It is advisable to apply several different media to improve the chances of isolation [89,94]. The variable results of *M. ovipneumoniae* cultivation and limited growth in mycoplasma media necessitated the development of a new growth medium free of ruminant proteins, named tryptone soy broth medium (TSB-1), which produced significantly greater and faster yields in comparison with some other media [90]. The advantages of using this medium were also observed in later studies by Jennings-Gaines et al. [95] and Maksimović et al. [20], especially when walled-bacterial inhibitors including amphotericin B, penicillin, and thallium acetate were added [95].

While intended for the growth of the caprine pathogen, *M. capricolum* subsp. *capripneumoniae*, Hayflick’s medium modified by Thiaucourt et al. [96] was shown to be successful for the cultivation of *M. ovipneumoniae* [20,30,35]; this is probably because this medium contains glucose and pyruvate, substrates which have been shown to be used by all strains so far tested [66]. Furthermore, incubating field samples for 48 h in modified TSB-1 at 37 °C and 10% CO_2_ greatly increased growth rates [95].

The growth of *M. ovipneumoniae* isolates has been shown to be very variable in vitro. Maksimović et al. [20] reported that the rapid growth phase varied between isolates from 24 h to 72 h, with most (60%) reaching a peak at 48 h. All strains were viable at 72 h after incubation, with declining viability seen at 96 h (13 of 20 remained viable; 65%), 120 h (9 of 20; 45%), and 144 h (4 of 20; 20%). A more rapid decrease in titer was observed after 72 h of incubation for goat strains when compared with sheep strains. Similarly, Wang et al. [21] reported that continuously passaging *M. ovipneumoniae* resulted in rapid cell death after a short stationary phase. To prolong the stationary phase, they proposed and validated a culture medium (Hayflick’s) with added serine.

It was also noted that *M. ovipneumoniae* isolates rarely reached visible turbidity or color change of the medium during the isolation procedure (personal observation). However, after subculture, the presence of the colonies on solid medium confirmed the growth in liquid medium. For a few isolates, color change of the phenol red indicator could be seen after approximately 5–6 days’ incubation when the cultures were most likely already in the late stationary or death phase (personal observation). Ackerman et al. [93] also reported senescence of cultures with rapid loss of cell viability after 5 days’ incubation. These observations indicated that the first three days of incubation are essential for efficient cultivation of most *M. ovipneumoniae* strains [20]. However, it should be noted that in some cases, the growth in broth was not visible. To avoid these false negative results, culturing for an average of 3 days in liquid medium followed by consecutive subcultures on solid media (known as a blind passage), and/or by polymerase chain reaction (PCR) test on the broth cultures should be applied [86,87].

Overall, the successful growth of *M. ovipneumoniae* required consideration of the inherent variability among strains, the time of subculturing and selection of appropriate media [20].

## 12. Detection and Identification

The identification of *M. ovipneumoniae* can be accomplished by conventional methods, including cultural examination, biochemical testing, serological tests like growth inhibition, and by molecular methods such as PCR [26].

Colonies of *M. ovipneumoniae*, grown on solid medium, containing the usual concentration of agar (1.5–2.0%), do not have the typical “fried egg” appearance because they lack the central mycoplasma downgrowth and can appear granular. Lower agar concentrations were shown to encourage the growth of typical colonies [22].

Immunocytohistochemistry has been used for the detection of the organism in diseased lungs [6,97]. Its advantage is that it is able to correlate antigen location with lung damage. However, more sensitive and specific methods for the detection of *M. ovipneumoniae* have been introduced in the last two decades. PCR identification of *M. ovipneumoniae*, based on the 16S RNA gene, was first developed in 2003 [98] and one decade later, a hsp70 gene-based PCR was reported with improved sensitivity and specificity [99]. PCR of the 16S rRNA gene followed by denaturing gradient gel electrophoresis (DGGE) enabled rapid detection and identification of mixed mycoplasma cultures isolated from multiple infections often seen in *M. ovipneumoniae* pneumonia. Commonly, *M. ovipneumoniae*/*M. arginini* and *M. ovipneumoniae/M. conjunctivae* may be found together and may be missed by conventional PCR [100]. Considering that many healthy animals carry low levels of *M. ovipneumoniae*, quantification of the number of organisms present in the samples is useful for determining whether the mycoplasma is having a major role in disease [26].

Both SYBR Green-based [101] and probe-based real-time PCR assays [11,102] have been designed for the rapid detection and quantification of *M. ovipneumoniae*. More recently, a real-time probe-based PCR for improved detection and differentiation of *M. ovipneumoniae* and a novel respiratory-associated *Mycoplasma* species (*M*. sp. nov.) was developed [103]. Loop-mediated isothermal amplification (LAMP) [104] and a biosensor comprising a manganese dioxide microsphere absorbing Cy5 labeled single strand DNA probe [105] have also been described before the sensitive and specific detection of *M. ovipneumoniae*. A newly developed isothermal recombinase polymerase amplification assay for the low-cost direct detection of *M. ovipneumoniae* in field conditions has been developed but requires further validation [106]. Matrix-assisted laser desorption ionization–time of flight mass spectrometry (MALDI-TOF MS) offers a rapid identification and differentiation of mycoplasma isolates, including *M. ovipneumoniae*, and supports the taxonomic resolution of animal mycoplasmas [107].

Newer technologies, such as next generation sequencing (NGS), have wide applications in well-equipped laboratories and can be used in epidemiological studies and the characterization of isolates. Although requiring bacterial isolates, WGS can provide simultaneous identification, typing and antimicrobial resistance prediction. In addition, it can be used for the study of virulence factors (genes), early detection of novel mechanisms of resistance, surveillance, and the rapid identification and tracking of infectious disease outbreaks [108]. As yet, there have been no reports of this technology being applied to *M. ovipneumoniae* research.

Serological testing can be useful for herd and flock surveillance of *M. ovipneumoniae* and detecting chronic and/or previously infected animals, even in the absence of active mycoplasma shedding [86]. The most widely used serological tests are enzyme linked immunosorbent assays (ELISA) [11,66,109] and indirect hemagglutination assays (IHA) [33,110,111]. However, serological testing can be unreliable because of the considerable antigenic heterogeneity of strains and variations in immune responses [66]. Low levels of antibodies to *M. ovipneumoniae* may be present during the early stages of infection, delaying the antibody response by 7–10 days after infection [86,112]. Increased levels of antibodies can be detected late in the clinical disease in animals that have apparently recovered from the clinical disease [112]. These findings indicate that serological testing is not reliable for detecting early stages of infection. The results of serology depend on the host response to infection and sampling time at different stages of disease [112]. Examining paired sera for a rising titer is recommended for estimation of an active *M. ovipneumoniae* infection [26]. Cross-reacting antigens between *M. ovipneumoniae* and other mycoplasma species should be assessed when developing serological tests [112,113]. Improved detection of highly variable antibodies to *M. ovipneumoniae* could be achieved by producing a cocktail of well conserved, highly expressed antigens in *M. ovipneumoniae* isolates.

## 13. Antimicrobial Susceptibility

The lack of internationally standardized guidelines for the antimicrobial susceptibility testing and interpretation criteria for animal mycoplasmas [114,115] confound evaluation of antibiotic resistance patterns of *M. ovipneumoniae* and therapy guidance. The careful and responsible use of antimicrobials might be considered after antimicrobial susceptibility testing (minimum inhibitory concentrations-MICs) has been performed and when other control measures have failed.

While mycoplasmas are inherently resistant to some antimicrobials, like the penicillins which target the cell wall, a few groups of antimicrobials have been shown to be effective in vitro against *M. ovipneumoniae*, including the fluoroquinolones, tetracyclines, and macrolides [8]. Furthermore, increasingly high MIC values have been reported for some oxytetracyclines [25] and macrolides in the last decade [2]. Although lincosamides, pleuromutilins, phenicols, and aminoglycosides are effective against mycoplasmas [115], higher MIC values have been reported for most of them when compared to other antimicrobials [8,19,24,25].

While antimicrobials may appear to be effective in vitro, their usefulness in the field is less certain [8]. Antimicrobials effective against mycoplasmas often produce a rapid respite, but animals with *M. ovipneumoniae* infections can often relapse and require further treatment [8]. Worryingly, systemic treatment of ewes with chronic infections with enrofloxacin, gamithromycin, tildipirosin, and tulathromycin was ineffective against *M. ovipneumoniae*, whereas combined systemic and intranasal enrofloxacin treatment eliminated *M. ovipneumoniae* [116]. Following subcutaneous administration, tulathromycin showed better therapeutic effectiveness in sheep affected by bacterial respiratory infection involving *M. ovipneumoniae* than other antibiotics tested, including gentamicin, oxytetracycline, thiamphenicol, tilmicosin, and enrofloxacin [117]. Interestingly, experimental infection of healthy domestic lambs with *M. ovipneumoniae* caused colonization of the upper airways that was resistant to antibiotic treatment [47].

Overall, the number of in vitro and in vivo studies is very limited to date, and more investigations are required before an optimum antibiotic regimen can be recommended. Given the absence of harmonized methods for testing and clinical breakpoints, MIC determination [114,115] is presently only a guide when selecting antibiotics appropriate for disease treatment and suppression of antimicrobial resistance.

## 14. Variations in Antimicrobial Susceptibility Profiles

In the absence of standardized procedures for MIC testing of antimicrobials against veterinary mycoplasmas and criteria for interpretation [114,115], differences in the antimicrobial sensitivity of different *M. ovipneumoniae* isolates remain uncertain. The development of epidemiological cut-off values (ECOFFs) (i.e., the highest MIC for organisms devoid of phenotypically detectable, acquired resistance mechanisms, which defines the upper end of the wild-type MIC distribution) is a necessary step when setting clinical breakpoints to guide therapy. The lack of ECOFFs prevents the separation of isolates with (non-wild-type) and without (wild-type) phenotypically detectable resistance, thus hampering surveillance and early warning of developing resistance [118,119].

Variability in strain susceptibility to antimicrobials complicates the control of *M. ovipneumoniae* infections [2]. Reported differences in MIC values for antimicrobial agents may be related to geographical origin and antimicrobial usage patterns, methodology, year of isolation, or ruminant host [2,8,19,24,25] (Table 3).

The highest MIC values have been seen with erythromycin (>1000 μg/mL) [24], spectinomycin and tylosin (128 μg/mL) [25], and streptomycin (>100 μg/mL) [24]. With a few exceptions, MIC values for the antimicrobials tested were mostly low for the isolates from France [19] and Bosnia and Herzegovina [25]. Fluoroquinolones appear to be the most effective antimicrobial agents against *M. ovipneumoniae* in vitro [8,19,25], but their use is strictly contra-indicated for domestic livestock because of their value in human health [120].

There have been few studies on *M. ovipneumoniae*, but those that have been conducted show that goat isolates were less sensitive to tylosin, enrofloxacin, norfloxacin, and oxytetracycline than sheep isolates [8,24,25]. Conversely, a more recent study [19] yielded slightly higher MIC values for florfenicol, tilmicosin, oxytetracycline, and spectinomycin for sheep isolates. The observed differences in antimicrobial susceptibility between sheep and goat isolates require further examination.

Variability in susceptibility of isolates to different antimicrobials, regional sensitivity to specific antimicrobials, variations in host ruminant species, the presence of multiple isolates, selection of antimicrobials, methodology of testing, and interpretation of the results should be carefully considered when testing in vitro.

## 15. Vaccines

Presently, there are no commercial vaccines available for *M. ovipneumoniae* pneumonia, probably because of the lack of awareness of the high prevalence and impact of this disease. The cost of producing a vaccine for small ruminants, many owned by poorer farmers grazing marginal land, is also of course one factor. Experimental vaccines have been reported with limited success, which may be due to the high immunological variability of *M. ovipneumoniae* strains, as well as its recently discovered intracellular nature [121]. Furthermore, immunity appears to be strain specific [87].

To date, research has shown the immune response to live and inactivated vaccines is low and variable and requires large amounts of protein to provoke one [122]. Moreover, few of the published studies have been without limitations, including experiments where the same strain was used for vaccination and challenge [121]. Several immunogenic proteins have been identified, including the membrane associated heat shock protein (HSB) 70 [83], but these are yet to be tested in animals.

The withdrawal of a *M. haemolytica*-vaccine in New Zealand a decade ago because of its ineffectiveness against ovine respiratory disease implicated *M. ovipneumoniae* as a major pathogen [8]. In the UK, outbreaks of pneumonia in *M. haemolytica*-vaccinated flocks, from which *M. ovipneumoniae* was a predominant finding, strongly suggested that a vaccine incorporating *M. haemolytica* and several representative strains of *M. ovipneumoniae* would, most likely, be beneficial to the small ruminant industry [8].

## 16. Control

In view of its largely unknown and variable impact and the occurrence of other priority diseases, national control authorities are unlikely to instigate eradication measures for *M. ovipneumoniae* pneumonia any time soon. Further surveys are clearly necessary to determine its true prevalence and economic impact. However, should these costs prove to be significant, then the following measures should be considered. In the light of *M. ovipneumoniae* heterogeneity, control of infections might be more effective by applying measures regionally and/or at the herd level. These include control of animal movements and isolation of new animals prior to introduction into herds, pre and post movement testing, isolation of clinically affected animals, the potential culling particularly of chronic carriers, monitoring herd health status, and good farming practice. Furthermore, all these measures would require testing strategies that could include improved culture methods coupled with serological testing for herd surveillance and application of new, rapid, and accurate molecular methods. In addition, vaccine production and serological tests should be based on antigenic characteristics of the isolates from individual flocks, while the choice of therapy needs to follow antimicrobial susceptibility patterns and current recommendations/restrictions for drug use in small ruminants. Wildlife protection is more challenging as it largely depends on exposure to domestic small ruminants that is not easily manageable. Lastly, control strategies would depend on their local feasibility, economic impact, and national disease control.

## 17. Conclusions

In the field, the number of reported cases of *M. ovipneumoniae* infections in domestic and wild ruminants appears to be rising. Whether this is due to a greater awareness of the disease following its detection in an increasing number of wild ruminant species or a real increase in cases in domestic species as a result of more testing is hard to tell at this stage. The diagnosis and control of the disease remain challenging and impaired by polymicrobial infections, the presence of *M. ovipneumoniae* in both affected and apparently healthy animals, as well as multiple heterologous strains within flocks, herds, or in single animals.

The major feature of *M. ovipneumoniae* is its high levels of genetic, biochemical, and phenotypic variability, which may be unique within the class *Mollicutes*. This may have arisen from a long evolutionary descent and close association with small ruminants, predating their domestication. The explanation for its heterogeneity may lie in its extreme ability, even by mollicute standards, for rapid evolution and/or adaptation to unpredictable environmental changes and distinct environments, including host species. This of course needs to be confirmed.

The pathogenicity of *M. ovipneumoniae* appears to have increased following spill-over into newly recognized host species when compared to its pathogenicity in its original hosts. Alternatively, these closely related ruminant species might have become more susceptible due to the better adaptation of *M. ovipneumoniae*. Its persistence and the (re) occurrence of severe outbreaks in domestic goats and wildlife, as well as the likelihood of a host shift and evolving adaptation strategies for increasing *M. ovipneumoniae* fitness, need to be further explored. Such variability provides a major challenge for control of infections.

## Figures and Tables

**Figure 1 pathogens-11-01477-f001:**
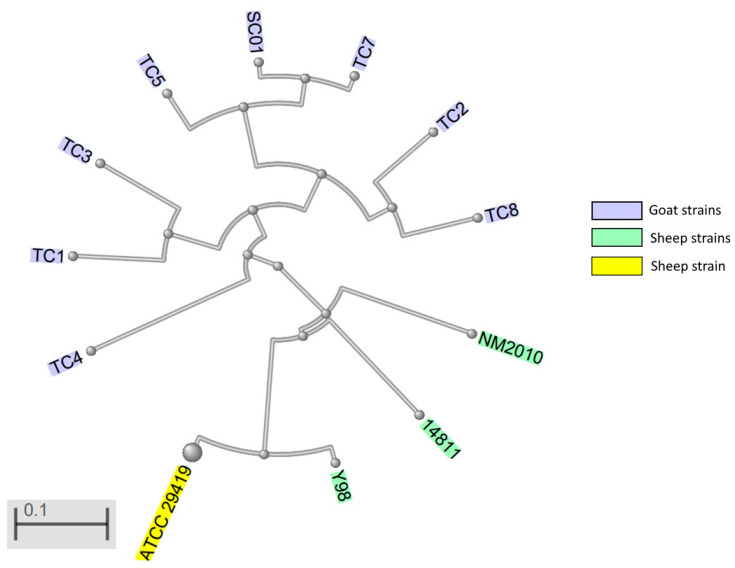
Whole-genome-based phylogenetic tree of *Mycoplasma ovipneumoniae* strains. TC1, TC2, TC3, TC4, TC5, TC7, TC8, SC01, goat strains, NM2010, sheep strain, China; 14811, sheep strain, France; Y98, type strain, sheep, Australia; ATCC 29419 type strain.

**Table 1 pathogens-11-01477-t001:** The host range of *Mycoplasma ovipneumoniae* in wild animals.

Animal	Geographic Location	Genus	Subfamily	Family	Ref.
Dall sheep (*Ovis dalli dalli*)	Toronto, Canada	*Ovis*	*Caprinae*	*Bovidae*	[9]
Bighorn sheep (*Ovis canadensis canadensis)*	Idaho, Oregon, Washington, USA	*Ovis*	*Caprinae*	*Bovidae*	[10]
Norwegian muskox (*Ovibos moschatus*)	Norway, Europe	*Ovibos*	*Caprinae*	*Bovidae*	[11]
Mountain goat (*Oreamnos americanus*)	Nevada, USA	*Oreamnos*	*Caprinae*	*Bovidae*	[14]
Desert Bighorn Sheep (*Ovis canadensis nelsoni*)	California, USA	*Ovis*	*Caprinae*	*Bovidae*	[15]
Moose (*Alces alces*)	Alaska, USA	*Alces*	*Capreolinae*	*Cervidae*	[12]
Caribou (*Rangifer tarandus*)	Alaska, USA	*Rangifer*	*Capreolinae*	*Cervidae*	[12]
Mule deer (*Odocoileus hemionus*)	Arizona, USA	*Odocoileus*	*Capreolinae*	*Cervidae*	[12]
White-tailed deer (*Odocoileus virginianus*)	Midwest region of the United States	*Odocoileus*	*Capreolinae*	*Cervidae*	[12]
Barren-ground caribou (*Rangifer tarandus granti*)	Alaska, USA	*Rangifer*	*Capreolinae*	*Cervidae*	[13]

**Table 2 pathogens-11-01477-t002:** Comparison of genome features of *Mycoplasma ovipneumoniae* strains.

*M. ovipneumoniae* Strain	SC01 (Goat, China)	TC 4 (Goat, China)	NM2010 (Sheep, China)	90 (Sheep, New Zealand)	14811 (Sheep, France)	ATCC 29419	Type Strain Y98	274 (Goat, B&H)
Accession No.	NZ_ AFHO01000010.1	NZ_ JOTH01000001.1	NZ_ JAKV01000001.1	NZ_ VZDP01000002.1	NZ_ JFAD00000000.1	NZ_ AGRE01000050.1	NZ_ KV765937.1	NZ_ CP079199.1
Genome size (bp)	1,020,601	996,705	1,084,159	1,031,345	1,071,500	1,020,200	1,038,480	1,081,404
GC content (%)	28.85	30.2	29.14	29	29.2	29.2	29.5	28.8
Number of genes (total)	750	792	772	771	769	788	817	778
Total number of CDSs	715	757	737	722	734	752	782	743
CDSs (with protein)	693	720	715	697	716	709	717	735
RNA genes	35	35	35	49	35	36	35	35
rRNAs	3	3	3	3	3	3	3	3
tRNAs	30	30	30	44	30	31	30	30
Other RNA	2	2	2	2	2	2	2	2
Pseudogenes	22	37	22	25	18	43	65	27

CDSs, coding sequences; GC, guanine-cytosine; B&H, Bosnia and Herzegovina.

**Table 3 pathogens-11-01477-t003:** Minimum inhibitory concentration range (μg/mL) of various antimicrobial agents for *Mycoplasma ovipneumoniae* isolates from sheep and goats.

Antimicrobial Agent		Animal (Number of Isolates Tested)	Geographical Origin and Year of Isolation	MIC Range (μg/mL)	Ref.
Fluoroquinolones	Enrofloxacin	Sheep (28)	France, 2007–2018	≤0.0625–0.125	[19]
		Sheep (12)	BH, 2009	<0.03–0.125	[25]
		Sheep (4)	UK, 2010	<0.12	[8]
		Sheep (27)	UK, 2003–2004	0.03–4.00	[2]
		Goat (28)	France, 2012–2017	≤0.0625–0.25	[19]
		Goat (12)	BH, 2010, 2011, 2015; Republic of Kosovo, 2008	<0.03–0.5	[25]
	Ciprofloxacin	Sheep (12)	BH, 2009	<0.03	
		Sheep (27)	UK, 2003–2004	<0.12–2	[2]
		Goat (12)	BH, 2010, 2011, 2015; Republic of Kosovo, 2008	<0.03–0.5	[25]
	Norfloxacin	Sheep (12)	BH, 2009	<0.03–0.25	
		Sheep (27)	UK, 2003–2004	<0.12–32	[2]
		Goat (12)	BH, 2010, 2011, 2015; Republic of Kosovo, 2008	0.0625–8	[25]
	Danofloxacin	Sheep (4)	UK, 2010	<0.12	[8]
		Sheep (27)	UK, 2003–2004	<0.06–8	[2]
	Marbofloxacin	Sheep (4)	UK, 2010	<0.12	[8]
Aminoglycosides	Gentamicin	Sheep (12)	BH, 2009	0.25–8	[25]
		Goat (12)	BH, 2010, 2011, 2015; Republic of Kosovo, 2008	0.125–2	
	Spectinomycin	Sheep (28)	France, 2007–2018	2–8	[19]
		Sheep (12)	BH, 2009	0.5–4	[25]
		Sheep (4)	UK, 2010	1–2	[8]
		Sheep (27)	UK, 2003–2004	0.5–64	[2]
		Goat (28)	France, 2012–2017	2–4	[19]
		Goat (12)	BH, 2010, 2011, 2015; Republic of Kosovo, 2008	0.5–128	[25]
	Streptomycin	Sheep (40)	New Zealand, 1975	6.2–>100	[24]
Tetracycline	Oxytetracycline	Sheep (28)	France, 2007–2018	≤0.0625–4	[19]
		Sheep (12)	BH, 2009	0.125–0.25	[25]
		Sheep (4)	UK, 2010	<0.12	[8]
		Sheep (27)	UK, 2003–2004	0.06–8	[2]
		Sheep (40)	New Zealand, 1975	0.25–1.5	[24]
		Goat (28)	France, 2012–2017	≤0.0625–1	[19]
		Goat (12)	BH, 2010, 2011, 2015; Republic of Kosovo, 2008	0.0625–32	[25]
Macrolide	Tylosin	Sheep (12)	BH, 2009	<0.03–0.25	
		Sheep (4)	UK, 2010	<0.12–0.25	[8]
		Sheep (27)	UK, 2003–2004	0.06–>64	[2]
		Sheep (40)	New Zealand, 1975	0.1–6.2	[24]
		Goat (12)	BH, 2010, 2011, 2015; Republic of Kosovo, 2008	<0.03–128	[25]
	Erythromycin	Sheep (4)	UK, 2010	0.25–1	[8]
		Sheep (27)	UK, 2003–2004	8–32	[2]
		Sheep (40)	New Zealand, 1975	62–>1000	[24]
	Tilmicosin	Sheep (28)	France, 2007–2018	2–32	[19]
		Sheep (4)	UK, 2010	1–2	[8]
		Sheep (27)	UK, 2003–2004	0.25–>64	[2]
		Goats (28)	France, 2012–2017	2–≥16	[19]
	Tulathromycin	Sheep (4)	UK, 2010	0.5	[8]
Phenicol	Florfenicol	Sheep (28)	France, 2007–2018	2–8	[19]
		Sheep (12)	BH, 2009	0.5–4	[25]
		Sheep (4)	UK, 2010	2–<32	[8]
		Sheep (27)	UK, 2003–2004	0.5–16	[2]
		Goat (28)	France, 2012–2017	2–8	[19]
		Goat (12)	BH, 2010, 2011, 2015; Republic of Kosovo, 2008	<0.03–8	[25]
	Chloramphenicol	Sheep (4)	UK, 2010	8	[8]
		Sheep (27)	UK, 2003–2004	1–32	[2]
		Sheep (40)	New Zealand, 1975	1.5–100	[24]
Pleuromutilin	Tiamulin	Sheep (12)	BH, 2009	0.125–0.5	[25]
		Goat (12)	BH, 2010, 2011, 2015; Republic of Kosovo, 2008	<0.03–0.5	
Lincosamide	Clindamycin	Sheep (27)	UK, 2003–2004	<0.12–2	[2]
	Lincomycin	Sheep (28)	France, 2007–2018	1–4	[19]
		Sheep (4)	UK, 2010	0.5	[8]
		Sheep (27)	UK, 2003–2004	0.25–8.00	[2]
		Goat (28)	France, 2012–2017	1–≥16	[19]
Aminocoumarins	Novobiocin	Sheep (40)	New Zealand, 1975	1.5–25	[24]

## Data Availability

Not applicable.

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
