# Peer review of "Mycoplasma ovipneumoniae: A Most Variable Pathogen"

_pathogens, 2022, doi:10.3390/pathogens11121477_

Round 1

Reviewer 1 Report

This is a very interesting review on Mycoplasma Ovipneumoniae.

In this review article the authors evaluated numerous pathogenicity mechanisms, including the formation of hydrogen peroxide, reactive oxygen species, and   toxins, have been postulated. It demonstrates broad metabolic activity in vitro, utilizing substrates such as glucose, pyruvate, and isopropanol; these patterns can be utilized to distinguish strains. Large differences in the susceptibility of strains to antimicrobials hinder the treatment of infections in the field, with many strains exhibiting high minimum inhibitory doses. The paper is well organized and written, however, there are few suggestions:

Minor concerns:

  1. The authors should increase the quality/size of figure 1 for better visualization.
  2. The authors should include few lines about the summary of the review in the abstract.

Author Response

Reviewer comment 1. The authors should increase the quality/size of figure 1 for better visualization.

Author response:  In accordance with your recommendation, we have made the correction of Figure 1.

Reviewer comment 2. The authors should include few lines about the summary of the review in the abstract.

Author response: In accordance with your recommendation, we have included the summary of the review in the abstract – “This review summarizes the current knowledge and identifies gaps in research on M. ovipneumoniae including its epidemiology in sheep and goats, pathology and clinical presentation, infection in wild ruminants, virulence factors, metabolism, comparative genomics, genotypic variability, phenotypic variability, evolutionary mechanisms, isolation and culture, detection and identification, antimicrobial susceptibility, variations in antimicrobial susceptibility profiles, vaccines and control.” (Lines 25-30).

Reviewer 2 Report

This excellent manuscript submitted by Maksimovic et al. describes the current knowledge of the main characteristics of Mycoplasma ovipneumoniae and its role in respiratory disease of small ruminants and emergence in wildlife species and identifies gaps in research and limitations regarding diagnosis and control of this increasingly important animal mycoplasma. The quality and comprehensiveness of the article is impressive; however, the authors may wish to consider these few minor modifications:

Title: Mycoplasma ovipneumoniae (in italics, ovipneumoniae in lower case)

Lines 134-176: It would be an asset to include information on the geographic location of these wild ruminants.

Line 582: … appears to have increased …

Author Response

Reviewer #2

 Reviewer comment 1. Title: Mycoplasma ovipneumoniae (in italics, ovipneumoniae in lower case)

Author response: In accordance with your instruction, we have corrected the title-

Mycoplasma ovipneumoniae: a Most Variable Pathogen (line 2).

Reviewer comment 2. Lines 134-176: It would be an asset to include information on the geographic location of these wild ruminants.

Author response: In accordance with your recommendation, we have added geographic location of the wild ruminants -Table 1 (line 178).

Reviewer comment 3. Line 582: … appears to have increased …

Author response: We have made the correction- appears to have increased (line 588).